# OpenReview forum: "Effectively Steer LLM To Follow Preference via Building Confident Directions"
_ICLR.cc/2025/Conference — Submitted to ICLR 2025_

### Official Review · Reviewer_rZ5M · 2024-10-31

**Soundness:** 2
**Presentation:** 3
**Contribution:** 2
**Rating:** 5
**Confidence:** 4

**Summary:**

This paper focuses on a crucial problem: how to efficiently achieve alignment between LLMs and human preferences and specifically targeted at activation modification in inference time. It also conducts a theoretical analysis of activation steering methods and proposes an approach to handle multiple preferences. Experimental results demonstrate that this method performs better than massive mean steering algorithms across several datasets and specific settings.

**Strengths:**

This paper addresses a key challenge: how to effectively align large language models (LLMs) with human preferences, with a particular focus on activation modification during inference.

The experiments are extensive in model architecture.

**Weaknesses:**

The main issue with this paper lies in its limited contribution and the vague description of its settings. For example, in line 97 where the contributions are outlined, first, the proposed theoretical framework seems to have very little connection to the presented method. Second, regarding the claim that it enables the alignment of multiple preferences, we could simply modify the output dimension of the linear probe in [1] from 2 to the number of preferences to achieve this expansion just like the authors did. As for "implicit model steering," [1] also operates in a similar way. Lastly, the "enabling adjustable levels of relevance" is actually controlled by a threshold, and in [1], this can similarly be achieved by controlling the number of selections in top-k merging, which is analogous to the method presented in this paper.

Regarding the setting and description, the descriptions of “user”, “user preference” $\theta$ , “user history” $\mathcal{H}$, history dataset $A$ are confusing and maybe redundant. Why we need to introduce the term “user history” to this context? The authors refer to news from different categories and responses generated by models with different labeled categories as the history data of different users, which is inappropriate and ambiguous. Why not just simply use “data of different preferences” to represent it? And also please provide a problem setup to make your setting more clearer.

The author does not validate the challenges they proposed, such as why existing methods would perform poorly when there is high noise in the samples, nor do they provide any experimental evidence to support this claim. If this challenge does indeed exist, the author does not explain why the method proposed in this paper is able to address it. In fact, the author mentions in the experimental section that the selection of the threshold also determines the level of introduced noise, which seems to contradict the claims made earlier.

Assumption 2 mentioned that ”$\hat{\theta}$ with the highest occurrence probability…”, but then P($\hat{\theta}$|…)<P($\theta$|…)for any $\theta$!=$\hat{\theta}$, which seems contradict the claims too.

The paper lacks detailed descriptions of the parameter settings for its method, raising concerns about transparency. On one hand, the proposed method involves a large number of hyperparameters that can be set, but the authors do not provide details about these settings. For instance, critical factors like layer l , the position of activation for the beginning token t, and the length s are not explained. The proposed method has too many hyperparameters, and the significance of these hyperparameters is questionable. Moreover, the authors do not offer a reasonable explanation or experimental validation for their settings.

Minor:
In line 466, the reference to figure 4 is incorrect. The authors should check the notations.

**Questions:**

In previous work [1], token-level activations were not concatenated, but this paper proposes concatenating activations. What is the motivation behind this? In fact, using the activation of the last token already represents the information of the entire prompt+response pair. Why is it necessary to concatenate a segment of token activations? Would it be possible to have ablation studies to verify the effect of this part?


Figure 3 is confusing, with some elements missing explanations. And, under "TVs," the value 0.45 > 0.4 is shown, but it seems the activation was not selected for averaging—why is this?


[1] Li, Kenneth, et al. "Inference-time intervention: Eliciting truthful answers from a language model." Advances in Neural Information Processing Systems 36 (2024).

---

> ### Author Response · Authors · 2024-11-22
> **Response to Reviewer rZ5M**
>
> >> The proposed theoretical framework seems to have very little connection to the presented method.
>
> **Response**: Thanks for your comment. We need to clarify that the theoretical framework is **closely related to the proposed method**. To see this, let us summarize the first two outlined contributions and corresponding sections here. (1) We propose a theoretical framework to quantify good steering directions that can effectively align the LLM output with user preference (Section 3), where the key is to find a steering vector that can approximately (not strictly) satisfy Assumption 1; (2) Guided by theory in (1), we design an algorithm to contruct an effective steering direction that approximately (maybe not strictly) satisfy Assumption 1 (Section 4).
>
> To be specific, in Section 3, we clearly illustrated that a good steering direction $v$ should lead to a high posterior probability of ground-truth preference $\theta^*$, i.e, $P(\theta^*|v)$ is close to $1$. In Section 4, **guided by the above theory**, we design algorithm to find a direction $v$ that can lead to high $P(\theta^*|v)$. Noticing that $P(\theta^*|v)\propto P(v|\theta^*)$, we train a classifier to estimate $P(v|\theta^*)$ contruct a confident direction that satisfies the theoretical requirment in Section (1).
>
>
>
> >>Regarding the claim that it enables the alignment of multiple preferences, we could simply modify the output dimension of the linear probe in [1] from 2 to the number of preferences to achieve this expansion just like the authors did. As for "implicit model steering," [1] also operates in a similar way.
>
> **Response**: Thanks for your comment. Our work already includes results from multi-dimensional linear probing on a **single fixed layer as a baseline** (black dash line in Fig.4 and Fig.5), which is slightly different from standard linear probing, to make the setting comparable to our method. Our proposed CONFST algorithm outperforms the linear probing in most cases.
>
> First, we need to clarify that the **original** linear probing method in [1] is **not directly comparable** to our work. The linear probe in [1] involves searching among all the layers and all the heads to find the most separable head activations, while our method only requires one fixed shallow layer activation. As we discussed in Section 2, linear probe method in [1] is **extremely expensive in computation and memory storage**, especially for large models. For example, linear probing in [1] needs to extract activation from $32$ layers, while our work only needs to extract $1$ layer. One of the purposes of this work is to find an **efficient** way to best align the model output with user preference. Thus, we could not directly extend the linear probe method to our setting.
>
>
> Second, our work already includes results from multi-dimensional linear probing on a single fixed layer **as a baseline**, but without searching across layers and heads. The result is that our method outperforms the mean steering method, as shown in Fig. 4, Fig. 5, and Fig. 10, which follow the approach of the Mass Mean Shift method from [1], but applied to **one fixed layer**. These results show that our CONFST algorithm outperforms this baseline, indicating that standard linear probing on multi-dimensions can be inadequate without extensive layer selection.
>
>
>
>
> >>The "enabling adjustable levels of relevance" is actually controlled by a threshold, and in [1], this can similarly be achieved by controlling the number of selections in top-k merging, which is analogous to the method presented in this paper.
>
> **Response**: We appreciate the reviewer's comment. While we acknowledge that our method involves top-k selection of directions, which shares similarities with the top-k merging approach in [1], we want to emphasize that our method is designed to be **more memory-efficient**. This efficiency is one of the **main objectives** of our algorithm design, and is one of the critical differences with [1]. As we discussed in Section 2, the method in [1] is **extremely expensive in computation and memory storage**, especially for large models. For example, method in [1] needs to extract activation from $32$ layers, while our work only needs to extract $1$ layer.

---

> > ### Author Response · Authors · 2024-11-22
> > **Response to Reviewer rZ5M(continue)**
> >
> > >>Redundant notations about user preference $\theta$, user history $\mathcal{H}$, history dataset $A$: Why we need to introduce the term “user history” to this context? Why not just simply use “data of different preferences” to represent it? And also please provide a problem setup to make your setting more clearer.
> >
> > **Response**: Thanks for your comment. We need to clarify that the introduction of "user history" is necessary, and cannot be replaced by  “data of different preferences”.
> >
> > To address this, let us first illustrate the realistic problem setup relevant to our approach. Consider a scenario where multiple users have been interacting with the same LLM model and have completed surveys provided by the LLM, indicating their preferred generated content in response to the same questions. Now, when one of these users submits a query, the LLM platform reviews the user’s past interaction history to learn their preferences. The model then generates an output aligned with the topic and style that this specific user prefers, ensuring personalized and contextually appropriate responses.
> >
> > In our setup, we emphasize that our method operates within a **realistic LLM-user interaction framework**, where the LLM only has access to the user's **past interaction history**. Unlike controlled experiments where "data of ground-truth" can be predefined and evaluated, this term is abstract and not directly applicable in real-world LLM-user interfaces.
> >
> > >>The author does not validate the challenges they proposed, such as why existing methods would perform poorly when there is high noise in the samples, nor do they provide any experimental evidence to support this claim. If this challenge does indeed exist, the author does not explain why the method proposed in this paper is able to address it. In fact, the author mentions in the experimental section that the selection of the threshold also determines the level of introduced noise, which seems to contradict the claims made earlier.
> >
> > **Response**:
> >
> > **High noise can lead to poor performance**: We have provided evidence that high noise can lead to poor performance by visualizing the activations of the AgNews dataset in **Fig. 11**. The left figure illustrates the scenario where direction selection was not performed. This results in extracted head activations containing irrelevant tokens, leading to noisy directions that are not aligned with the intended topics. These exctrated noisy tokens are used to construct a steering vector in the mean steering method [1], as a consequence, the baseline (black dash line) in Fig.4 and Fig.5 uses the mean steering (or Massive Mean Shift) method from [1] without direction selection, which is worse than our purposed method. We have also added two additional baselines Activation Addition and In-context Learning during rebuttal phase, and our proposed method outperforms these two methods as well.
> >
> > **How our method can deal with noisy data**: We have provided a visualization of the confident head activations in the right figure of Fig. 11. Compared to the left figure, the four types of news topics in the right figure are more distinctly separated. This makes it intuitively evident that the steering direction for aligning the model output is more clear. This enhanced separation demonstrates the effectiveness of our method in identifying the most relevant activations and constructing precise steering directions to align outputs with user preferences.
> >
> > **Relationship between threshold and noise**: We clarify that there is **no contradiction** between Claim 1 and the discussion about the selection threshold and noise level in the experiment section. In fact, our experimental results **support** Claim 1. In our Claim 1, we characterize a good steering direction $v$ by evaluating the posterior probability of latent preference, i.e, $P(\theta^*|v)$. A high $P(\theta^*|v)$ indicates  that the model output can be effectively aligned with user preferences using the steering direction $v$. In our experiments, setting **a higher confidence threshold helps filter out noise**, leading to more reliable steering directions that better represent the ground-truth preference and improve performance. This is evident in straightforward steering tasks, such as aligning with science/tech or world topics in the AgNews dataset. However, in more challenging tasks, **a high threshold may result in fewer selected directions**. While these directions are representative of the preference, the constructed $v$ from averaging them may be **biased** from the true $\theta^*$ due to an **insufficient number of samples**. In such cases, $P(\theta^*|v)$ is low, reflecting the potential degrade in steering performance.

---

> > > ### Author Response · Authors · 2024-11-22
> > > **Response to Reviewer rZ5M(continue)**
> > >
> > > >>Assumption 2 mentioned that $\widehat{\theta}$ with highest occurence probability, but then $P(\widehat{\theta}|...)<P({\theta}|...)$.
> > >
> > > **Response**: Thank you for pointing out the issue. This is a typo and we have fixed it in the revised version. It should be $P(\widehat{\theta}|...)>P({\theta}|...)$
> > >
> > > >>The paper lacks detailed descriptions of the parameter settings for its method, raising concerns about transparency.
> > >
> > > **Response**: We have attached the table including all the parameters in Appendix E.
> > >
> > > >>Question about concantenation of activations.
> > >
> > > **Response**: In some of our experiments, we use the concatenation of the last four tokens. This approach provides more informative and distinguishable activations compared to using only the last token. In the Emotion dataset processed with GPT-2-XL, when we use the concatenated activations of the last four tokens, the activationa become more distinct and informative, enhancing the separation of different classes or user preferences. When we use 4 tokens, the classification accuracy is 40%, while using last one token is 37%. Please see **Appendix E**.
> > >
> > >
> > > >>Figure 3 is confusing, with some elements missing explanations. And, under "TVs," the value 0.45 > 0.4 is shown, but it seems the activation was not selected for averaging—why is this?
> > >
> > > **Response**: Thank you for pointing out the issue. The word should be selected in this case. We have revised the figure in the updated version and change the threshold $\beta=0.5$ in the revised version.
> > >
> > > [1] Li, Kenneth, et al. "Inference-time intervention: Eliciting truthful answers from a language model." Advances in Neural Information Processing Systems 36 (2024).

---

> ### Author Response · Authors · 2024-12-02
>
> Dear Reviewer rZ5M,
>
> Thank you for your comment and suggestions. Since the discussion period is ending on Dec 2nd, we kindly ask you to review our response to your concerns. We hope that our response could solve your questions, and we would greatly appreciate it if you could reconsider your score. Thank you again for your time!
>
> Authors

---

> > ### Author Response · Authors · 2024-12-03
> >
> > Dear Reviewer rZ5M,
> >
> > Thank you for your comment and suggestions. Since the discussion period is ending on Dec 2nd, we kindly ask you to review our response to your concerns. We hope that our response could solve your questions, and we would greatly appreciate it if you could reconsider your score. Thank you again for your time!
> >
> > Authors

---

### Official Review · Reviewer_wHLb · 2024-10-31

**Soundness:** 3
**Presentation:** 2
**Contribution:** 3
**Rating:** 5
**Confidence:** 4

**Summary:**

This paper proposes an effective theoretical framework, as well as a general and efficient preference alignment method that guides large language models (LLMs) to generate outputs that match user preferences. The paper gives the corresponding theoretical analysis on generating derivation. The proposed method CONFST achieves personalized alignment by modifying model activations without requiring fine-tuning, and the authors validated the robustness of CONFST on several large foundation models.

This excellent work has brought new insights to the community in terms of theoretical contributions and preference alignment schemes. However, there is a lack of consideration in terms of experimental design and the design of multi-user preference conflicts.

**Strengths:**

The paper provides an innovative theoretical framework that deepens the understanding of preference alignment principles.

CONFST is general and efficient, allowing alignment of multiple user preferences without fine-tuning, thus reducing resource costs.

The robustness and broad applicability of the algorithm are demonstrated on several benchmark models, including GPT-2 XL, Mistral, and Gemma.

**Weaknesses:**

The experimental setup is relatively simple; for instance, the topic shift task is primarily a sentiment classification task, a classical one that even traditional ML methods would be easy to produce a good performance, which somewhat limits the persuasiveness of this work. More kinds of evaluations may help.

The baseline comparisons are limited. A dedicated section explaining the absence of other baselines, or additional baseline comparisons, would strengthen the paper.

The paper lacks case studies, which would demonstrate the method’s performance in specific contexts. Personalized alignment could benefit from concrete case studies to showcase detailed performance.

The paper does not consider potential conflicts in multi-user preference alignment. Since CONFST supports multiple preference alignments, how would conflicting preferences among users affect model output? It seems that this paper does not address this type of conflict in its design.

**Questions:**

How do the authors demonstrate that the method can align with a broader range of unexplored preferences?

If conflicting user preferences are input, how would this affect the model's output?

If the authors can effectively address my concerns or clarify my potential misunderstandings through experiments or discussion (supporting material should be included), I would be happy to raise my score (Please see "Weaknesses" and "Questions" for detailed feedback).

---

> ### Author Response · Authors · 2024-11-22
> **Response to Reviewer wHLb**
>
> >>Weakness 1: The experimental setup is relatively simple; for instance, the topic shift task is primarily a sentiment classification task, a classical one that even traditional ML methods would be easy to produce a good performance, which somewhat limits the persuasiveness of this work. More kinds of evaluations may help.
>
> **Response**: Thank you for your comment. First, we need to clarify that the main contribution of our work is to **steering the model to generate content aligned with user preferences** (i.e., generative tasks), rather than performing traditional sentiment classification. Although we utilize classic classification datasets like AgNews and Emotion in our experiments, our goal is to guide the model’s output to align with specific user preferences, represented by classes in these datasets—a task that traditional ML methods are not designed to accomplish. Employing the aforementioned dataset for generative tasks has been found at [1], which uses Emotion dataset. In addition to these topic shift tasks, we also use more complicated NLP datasets, e.g, HelpSteer, oasst2 to perform more challenging pattern shift alignment tasks.
>
> Second, we additionally provide other evaluation related to topic shifting and other style shifting, covering **conciseness, helpfulness, humor, and detoxification directions**. In Fig. 8, we show the classification of four types of news using an **LLM auto-evaluation** (GPT-4o-mini), which is more robust and accurate than conventional classification models.
>
> >>Weakness 2: The baseline comparisons are limited. A dedicated section explaining the absence of other baselines, or additional baseline comparisons, would strengthen the paper.
>
> **Response**: Thanks for your suggestion. During the rebuttal phase, we added results for additional baselines, e.g., Activation Addition (AcdAdd) and In-context Learning (ICL) in addition to the mean steering method included in the original paper. However, it is important to highlight that these methods are **not directly comparable** to ours. As discussed in the related works section, methods like AcdAdd and ICL require the explicit expression of the target direction, however, our approach is based on implicit steering derived from user history and preferences. This fundamental difference in how user preferences are inferred and applied distinguishes our method from these baselines.
>
> Despite the difference in the setting, we still provide the updated Fig.4 and Fig.5 in the revised version with addtional baselines AcdAdd and ICL with two context examples. Notice that the ActAdd method is slightly different from the original paper [2], since the original paper uses OpenWebText as retrieval database, which is different from our user interface setting. Instead, we simply steer the model with activation of summarized explicit alignment direction, e.g activation of word "science" for AgNews, or activation of word "joy" for Emotion. From the result, we can conclude that AcdAdd can fail due to the summary of preference is not accurate, and without the retrieval database OpenWebText used in [2], AcdAdd cannot be effective. ICL has better and more stable performance than AcdAdd, but still falls behind our method due to the bias in inferring preference with limitation of contexts.
>
> >>Weakness 3: The paper lacks case studies, which would demonstrate the method’s performance in specific contexts. Personalized alignment could benefit from concrete case studies to showcase detailed performance.
>
> **Response**: Thanks for your comment. We have attached some examples of case study in **Appendix D**. We consider the preference of four types of news in Fig. 12 and show an example of generated steered output. In Fig.13, we show the example output of steering towards different styles/patterns. We understand this case study is important, we have to put it in the appendix due to the limited space; we will move it to main text in the final version.

---

> > ### Author Response · Authors · 2024-11-22
> > **Response to Reviewer wHLb (continue)**
> >
> > >>Weakness 4: The paper does not consider potential conflicts in multi-user preference alignment. Since CONFST supports multiple preference alignments, how would conflicting preferences among users affect model output? It seems that this paper does not address this type of conflict in its design.
> >
> > **Response**: Thanks for your comment. From our understanding, the conflicting preference has two potential meanings: (a) Different and conflicting preferences **between different users**; (b) Conflicting preferences **within one user**. We will address both potential meaning since we are not sure which the reviewer refers to.
> >
> >
> > **(a) Different and conflicting preferences between different users**: In this case, we deal with the case where users have totally different steering directions. For example, one user expects the response from LLM to be concise, while another user expect it to be detailed and long. In fact, this case is exactly what our method trying to deal with. Our proposed CONFST algorithm can identify these two different and conflicting preferences, and construct two confident steering directions that represent each preference respectively. In the inference stage, the algorithm steers towards the corresponding direction based on each user's preference. Each user will their own vectors. The vectors will only be added to LLM during inference.
> >
> > **(b) Conflicting preferences within one user**: In cases where a user has multiple preferences combined into one, the realistic setting is that there are no conflict within a single style. For example, a user typically has only one preference for a specific style, so conflicting preferences (e.g., expecting both long and short responses) do not arise. However, we do support combining different steering directions for different styles, which generally do not conflict. For instance, a user may want the content to be both helpful and concise, which is practical and non-contradictory. The combination of such steering directions is supported, as shown in our experimental results in Fig. 7, when we steer towards both helpfulness and conciseness direction, the response is shorter and more helpful.

---

> > > ### Author Response · Authors · 2024-11-22
> > > **Response to Reviewer wHLb (continue)**
> > >
> > > >>Question 1: How do the authors demonstrate that the method can align with a broader range of unexplored preferences?
> > >
> > > **Response**:
> > >
> > > **Unexplored preferences**: We emphasize that our proposed method can be applied to align with **any unexplored preference**. Our method is not dependent on whether current preference has been seen before or not, instead, given any new user's interactive history with LLM, we can always extract a quantity to represent the new user's preference together with the users that have seen before. Suppose we have some users and they have some interactive history with LLM. To align towards their preference, we simply follow the steps in our proposed CONFST algorithm: First, get the activations from the new users' history, and combine it with the old users' history. Then classify across all the users. Next, we select the confident directions that can represent each of the new user. Finally a steering direction can be built to steer the LLM output towards the preference direction.
> > >
> > > **Broader preferences**: Although our experiments have not shown results on aligning towards more user preferences, our proposed method has shown a better capability of scaling up compared to other methods. For examples, in Fig.4 and Fig.5, when we perform **multiple topic shift** on one fixed layer activation, our method **outperforms** the mean steering method. The reason behind this is that, our method is better in **ditinguishing multiple different preferences**. Evidence can be found in Appendix C Fig.11 where there are $4$ alignmenmt options. The left side is the original PCA plot of activations, and the right side is the PCA plot of selected confident directions. The mean steering method constructs the steering vector on mean activation of each type of news, while our method constructs the steering vector based on **the mean of selected activations**. The selection of confident directions makes the four types of news **more separable**, indicating a more clear steering direction.
> > >
> > > In terms of broader range of preferences, we believe our method can be extended to accommodate more preferences than the current experiments. Our method can be scaled up to a wider range of user preferences using the style embedding model proposed in [3]. In existing literature, some works address LLM personalization for hundreds of user preferences through Retrieval Augmented Generation (RAG) [4], where user preferences are represented using embedding models from [3].  These methods have shown promising results in aligning model outputs with diverse user needs. Similarly, with a more accurate and comprehensive embedding representation, our proposed CONFST algorithm holds significant potential to scale up and cater to more complex and unexplored user preferences, further enhancing personalization capabilities.
> > >
> > >
> > >
> > > >>Question 2: If conflicting user preferences are input, how would this affect the model's output?
> > >
> > > **Response**: Please refer to Weakness 4.
> > >
> > > Reference:
> > >
> > > [1] Konen, Kai, et al. "Style Vectors for Steering Generative Large Language Model." arXiv preprint arXiv:2402.01618 (2024).
> > >
> > > [2] Turner, Alexander Matt, et al. "Activation addition: Steering language models without optimization." arXiv e-prints (2023): arXiv-2308.
> > >
> > > [3] Wegmann, Anna, Marijn Schraagen, and Dong Nguyen. "Same author or just same topic? towards content-independent style representations." arXiv preprint arXiv:2204.04907 (2022).
> > >
> > > [4] Neelakanteswara, Abhiman, Shreyas Chaudhari, and Hamed Zamani. "RAGs to Style: Personalizing LLMs with Style Embeddings." Proceedings of the 1st Workshop on Personalization of Generative AI Systems (PERSONALIZE 2024). 2024.

---

> > > > ### Comment · Reviewer_wHLb · 2024-11-22
> > > >
> > > > Thanks for your response to all the issues raised.
> > > >
> > > > Regarding Q1, could you provide some additional experimental results?

---

> > > > > ### Author Response · Authors · 2024-11-24
> > > > > **Additional experiment about Q1 in Fig. 18 and Fig.19**
> > > > >
> > > > > **Response**: We have provided additional experiments regarding more **unexplored** preference in Fig.18 and Fig.19, and the result shows that our method has good performance on new unexplored preference.
> > > > >
> > > > > As we reply in the response, to align towards new user's unexplored preference, we simply follow the steps in our proposed CONFST algorithm: First, get the activations from the new users' history, and combine it with the old users' history. Then classify across all the users. Next, we select the confident directions that can represent each of the new user. Finally a steering direction can be built to steer the LLM output towards the preference direction.
> > > > >
> > > > > We set the four types of news in AgNews dataset as four current preferences, and generate additional data about "movie" as a new user preference. We use our proposed CONFST algorithm to align the new user's preference about movie related topic. We set the threshold $\beta=0.3,0.4,0.5,0.6$. For evaluation, we use GPT-4o-mini API with a given prompt: "You are a classification assistant for checking the type of the news. We would like to request \
> > > > > your feedback on the content. Choose from the following four options: A. world news; B. sports news; C. business news\
> > > > > D. science and tech news. E. movie/story/creation. Output A,B,C,D,E only."
> > > > >
> > > > > Additionally, we provide the PCA plot in Fig. 19, which is similar to Fig.11 (Please refer to the previous response for detailed discussion of Fig.11) The left figure in Fig. 19 shows the visualization of all the samples without selection, where the four old preferences and the new preference cannot be identified clearly. While in the right figure, when selection is performed, the new preference, which is about movie topic, can be clearly separated from other preferenences. This result indicates that our proposed method is applicable to align unexplored preferences.
> > > > >
> > > > > Due to the time limit of rebuttal, we are not able to use the embedding method from [1] to scale up our experiment to LLM personalization task, which requires heavy coding. We will leave this to our future work.
> > > > >
> > > > > Reference:
> > > > >
> > > > > [1] Wegmann, Anna, Marijn Schraagen, and Dong Nguyen. "Same author or just same topic? towards content-independent style representations." arXiv preprint arXiv:2204.04907 (2022).

---

> ### Author Response · Authors · 2024-12-02
>
> Dear Reviewer wHLb,
>
> Thank you for your comment and suggestions. Since the discussion period is ending on Dec 2nd, we kindly ask you to review our response to your concerns. We hope that our response could solve your questions, and we would greatly appreciate it if you could reconsider your score. Thank you again for your time!
>
> Authors

---

> > ### Author Response · Authors · 2024-12-03
> >
> > Dear Reviewer wHLb,
> >
> > Thank you for your comment and suggestions. Since the discussion period is ending on Dec 2nd, we kindly ask you to review our response to your concerns. We hope that our response could solve your questions, and we would greatly appreciate it if you could reconsider your score. Thank you again for your time!
> >
> > Authors

---

### Official Review · Reviewer_XAGT · 2024-11-01

**Soundness:** 2
**Presentation:** 1
**Contribution:** 2
**Rating:** 6
**Confidence:** 3

**Summary:**

This paper proposes a theory to explain how steering vectors work, and describes an algorithm for constructing steering vectors inspired by this theory. The paper then empirically tests the proposed steering method across different topics and datasets.

**Strengths:**

Originality: To my knowledge, this method is original, however I have a limited background in this area.
Clarity: Apart from some minor typos, the paper made sense overall.

**Weaknesses:**

Significance: My main issue with this paper is one of framing -- while the paper claims to be about steering LLMs to follow a user's preferences, it is primarily about topic modeling and steering an LLM to generate text which is on a particular topic.
Quality:
2a) The theory seems to obscure some questions around how to infer the user preferences $\theta$ given the user history $(A_{1:n}$. Namely, the steering vector clearly influences the probability of a particular output $A_i$ via $P_M(A_i | \theta; X )$, but the proposed theory simply assumes that $\theta$ can be directly estimated from $f(\Tau(A_{1:n}))$, and looks at the Bayesian posterior over $\theta^*$  given $f(\Tau(A_{1:n}))$ without proposing a concrete likelihood function $Pr(f(\Tau(A_{1:n})) | \theta))$. Instead, they implicitly assume that logistic regression trained to separate between $f(\Tau(A_{1:n})) | \theta)$ for different topics will work.
2b) The paper does not compare its results to a baseline method such as the original Activation Addition method by Turner et al, making it unclear that the proposed method beats the prior art.

**Questions:**

1) Does the proposed method outperform any baselines?
2) Is there empirical evidence to suggest that the optimal steering vector $\theta$ can be inferred solely from the activations $f(\Tau(A_{1:n}))$?
3) What is the likelihood function for $Pr(f(\Tau(A_{1:n})) | \theta))$?

---

> ### Author Response · Authors · 2024-11-22
> **Response to Reviewer XAGT**
>
> >>Weakness regarding significance:  My main issue with this paper is one of framing -- while the paper claims to be about steering LLMs to follow a user's preferences, it is primarily about topic modeling and steering an LLM to generate text which is on a particular topic.
>
> **Response**: Thanks for your comment. Let us first  clarify that there is misunderstanding about our work. Our model steering approach goes beyond simple topic modeling; it also focuses on adjusting the output to adhere to **specific language styles and patterns**.  As we will argue shortly,  both our technical approach as well as a number of numerical experiment demonstrate that our approach goes much beyond simple topic modeling.
>
> First, from technical point of view, our proposed CONFST could **effectively extract the user's preference of style or pattern**, in addition to the interested topic. To be specific, we use the conciseness steering as an example. We provide an additional visualization in the form of a PCA plot, illustrating the activations of the last token for concise responses versus long responses in **Fig.12 Appendix C**. We can conclude from Fig. 12 that we can clearly see the **difference between concise and long responses**, while our CONFST algorithm selects the circled confident conciseness directions to construct steering vector. These selected directions typically correspond to responses with a simple "Yes" or "No" for yes/no questions, or a single word for "what" questions. This is different from topic modeling that extract the **most common words** that align with certain interested topic, since our method in style/pattern shifting captures the **response format** instead of key words that can align with user's preferred style.
>
>
> Second, our experiments also demonstrated that our proposed CONFST algorithm could effectively perform the style/pattern alignment. We will discuss our result in the following
>
> **Conciseness**: Fig. 6 and Fig.8 show the result for conciseness steering. With our CONFST algorithm, the word counts after steering on both Mistral and Gemma model decreases significantly.
>
> **Helpfulness**: Fig.7 includes the result for the helpfulness steering direction. After steering, the generated response is evaluated to have higher helpfulness score by GPT-4o-mini.
>
> **Humor**: Fig.9 shows the result for the humor direction steering when the LLM generates response to a prompt that demonstrates dissatisfaction. GPT-4o-mini votes the steered generated outputs to be more indirect.
>
> **Detoxification**: Fig. 10 shows the result for detoxification, which evaluates the LLM output responding to prompts that can potentially lead to toxic content. Our CONFST outperforms the Massive Mean Shift method baseline, indicating our method ensures a more appropriate and safe response.

---

> > ### Author Response · Authors · 2024-11-22
> > **Response to Reviewer XAGT (continue)**
> >
> > >>Quality 2(a): Assumes $\theta$ can be totally inferred from $f(\mathcal{T}(A_{1:n}))$ without concrete likelihood function $P_r(f(\mathcal{T}(A_{1:n}))|\theta)$; use of logistic regression.
> >
> > **Response**: Thanks for your concern. We will address your comments in our **theory and algorithm** parts respectively:
> >
> > **Theory**:  Our Claim 1 and Claim 2 illustrate that when the groud-truth preference $\theta^*$ can be accurately inferred from the quantity $f(\mathcal{T}(A_{1:n}))$ of user history (Assumption 1), the generated output of LLM is close to the user's ground-truth preferred content. In our theory, 1. We do assume the user preference can be totally inferred from user's history (Assumption 1), and we will argue below that  this assumption is **reasonable**; 2. The concrete likelihood function $P_r(f(\mathcal{T}(A_{1:n}))|\theta)$ is not required in our theory.
> >
> > First, we clarify that this assumption is **reasonable**, at least conceptually. In Assumption 1, we assume that
> > \begin{align*}
> > \limsup_{n\rightarrow\infty}{P\left(\theta^* \mid f\left(\mathcal{T}(A_{1:n})\right)\right)} = 1,\;\forall \theta\neq\theta^*.
> > \end{align*}
> >
> > In this assumption, the accurate inference of  $\theta^*$ can be achieved only when the number of user samples is **infinity**. This can be understood similarly to the Law of Large Numbers (LLN), where the function $f$ in LLN is the average function, function $\mathcal{T}$ in LLN is the identity function. The true expectation of a parameter can only be accurately estimated by the averaged samples with probability $1$ when an **infinite number of samples** is available. In our Assumption 1, $\theta^*$ is a latent and abstract quantity to estimated that represents the ground-truth preference, $\mathcal{T}$ is the feature extraction function, while the function $f$ can be more complicated since it can be any valid function and there is no requirement on specific formula of $f$. Similar to LLN, the ground-truth quantity $\theta^*$ can only be estimated when features of **infinity samples** are provided.
> >
> >
> > Second, the Assumption 1 is abstract and **does not require** any explicit formula of likelihood function $P_r(f(\mathcal{T}(A_{1:n}))|\theta)$. Our Claim 1 and Claim 2 work for any valid likelihood function, though the explit likelihood function will be needed in finding the steering vector with high likelihood $P(v|\theta^*)$.
> >
> > Similar setting can be found in literature. For example, in [1] equation (94) it is comcluded that latent concept can be inferred from infinity long contexts, and concrete posterior distribution of latent concept conditioned on context is also not defined.
> >
> > **Algorithm**: As discussed in theory part, the explicit formula of $P_r(f(\mathcal{T}(A_{1:n}))|\theta)$ is required to find the steering vector $v$ that satisfy Assumption 1 (not strictly),
> > \begin{align*}
> > \limsup_{n\rightarrow\infty}{P\left(\theta^* \mid f\left(\mathcal{T}(A_{1:n})\right)\right)} = 1,\;\forall \theta\neq\theta^*.
> > \end{align*}
> >
> > However, since the likelihood function is abstract and can vary across different datasets, it's diffcult to estimate it. A practical method is to use logistic regression method instead to find the confident direction $v$ with high likelihood function $P(v|\theta)$. The underlying distribution $P(\theta|v)$ that logistic regression aims to approximate is the Bernoulli distribution. As a concrete example, we consider the AgNews dataset. For any fixed $v$, logistic regression approximates the assumed Bernoulli distribution $P('world'|v)=p_1,P('sports'|v)=p_2$,$P('business'|v)=p_3,P('science/tech'|v)=p_4$. If $v$ can higly represent 'science/tech', then $p_4>>p_1,p_2,p_3$. Logistic regression can effectively estimate a high $p_4$ is $v$ is a typical representation of 'science/tech'.
> >
> >
> > In fact, this method is able to find the confident directions effectively. In Fig.11, the left side is the original PCA plot of activations, and the right side is the PCA plot of confident directions selected by logistic regression. As shown in the right figure, the logistic regression model can **effectively** select the confident directions and makes the four types of news are more separable and well-clustered, demonstrating that logistic regression can identify directions that align with user preferences.

---

> > > ### Author Response · Authors · 2024-11-22
> > > **Response to Reviewer XAGT (continue)**
> > >
> > > >>Quality 2(b): The paper does not compare its results to a baseline method such as the original Activation Addition method by Turner et al, making it unclear that the proposed method beats the prior art
> > >
> > > **Response**: First, we need to clarify that the Activation Addition (ActAdd) method is **not directly comparable** to our experiment setting. The reason is that our method aims to **implicitly** steer the model output based on the user history, while the explicit expression of target direction is not available.  For example, if our goal is to generate content about science, the algorithm should be able to capture user's interest in science and generate related content However, ActAdd requires **explicit** expression of alignment direction, in the same example, key words "science", "biophysics" are required.
> > >
> > > Despite the difference in the setting, we provide the updated Fig.4 and Fig.5 in the revised version with addtional baselines ActAdd and ICL with two context examples. Notice that the ActAdd method is slightly different from the original paper [4], since the original paper uses OpenWebText as retrieval database, which is different from our user interface setting. Instead, we simply steer the model with activation of summarized explicit alignment direction, e.g activation of word "science" for AgNews, or activation of word "joy" for Emotion. From the result, we can conclude that ActAdd can fail due to the summary of preference is not accurate, and without the retrieval database OpenWebText used in [4], ActAdd cannot be effective. ICL has better and more stable performance than ActAdd, but still falls behind our method due to the bias in inferring preference with limitation of contexts.
> > >
> > >
> > >
> > > >>Question 1: Does the proposed method outperform any baselines?
> > >
> > > **Response**: We have provided the result of mean steering [2][3] in Fig.4, Fig.5 and  Fig.10 of original version, though the mean steering method used in our baseline is slightly different from literature to make it comparable to our setting, which will be discussed in the next paragraph. In all the above experiments, our method **outperforms** baseline steering methods. Additionally, we have added **two more baselines** in Fig.4 and Fig.5 of the revised version. Please refer to the response to Quality 2(b) for details.
> > >
> > > Next, let us briefly summarize the differences between the settings of mean steering in our paper and those in the existing literature.
> > > First, in literature, mean steering method (or Massive Mean Shift) uses the difference between the mean activations of target direction and the other unwanted direction as the steering vector. The mean steering method in our baseline uses the mean activations of target user as steering vector without subracting the unwanted direction, since we have multiple unwanted directions. Second, we do not perform the layer selection procedure in [1], since one of the purposes of this work is to find an efficient way to best align the model output with user preference. In [1], it is required to select $32$ layer activations from the Mistral model to select the most separable activations, which is extremely inefficient. While our work only needs to extract $1$ fixed layer activation.

---

> > > > ### Author Response · Authors · 2024-11-22
> > > > **Response to Reviewer XAGT (continue)**
> > > >
> > > > >>Question 2: Is there empirical evidence to suggest that the optimal steering vector $\theta$ can be inferred from solely from the activations $f(\mathcal{T}(A_{1:n}))$?
> > > >
> > > > **Response**:  We presented a visualization of activations for different preferences in **Fig. 11**, demonstrating that, when confident steering directions are obtained, the differences between preferences can be **clearly captured and visualized**. This result shows that preferences can be inferred and distinguished as distinct clusters.
> > > > On the right side of Fig.11, we extract the first layer activations (extraction function is $\mathcal{T}$) from GPT2-XL of each AgNews sample $A_i$, use PCA analysis to choose the first and second principal direction, and plot it on the Fig.11. We define the function
> > > > $f$ to select the confident directions for each type and compute the average of the selected directions. The figure on the right shows the four types of news can be separated clearly, and the averaged selected activations $f(\mathcal{T}(A_{1:n}))$ of each news type are also separated. A similar visualization is shown in Fig. 2(B) of [2], where the "truthful" preference is illustrated using a PCA plot. However, as discussed in Weakness 2, since user preference is latent, so we cannot provide a **more direct** evidence that it can be fully inferred from steering directions.
> > > >
> > > > >>Question 3: What is the likelihood function for $P_r(f(\mathcal{T}(A_{1:n})))$?
> > > >
> > > > **Response**: As we discussed in Weakness 2, the explicit likelihood function is not required in our theory, but is required in our algorithm to find steering direction $v$ that aligns with user preference. However, this likelihood function is abstract and can vary across datasets, and it is difficult to estimate, so we avoid using the explicit likelihood function to find confident $v$.
> > > >
> > > >
> > > > Reference:
> > > > [1] Xie, Sang Michael, et al. "An explanation of in-context learning as implicit bayesian inference." arXiv preprint arXiv:2111.02080 (2021).
> > > >
> > > > [2] Li, Kenneth, et al. "Inference-time intervention: Eliciting truthful answers from a language model." Advances in Neural Information Processing Systems 36 (2024).
> > > >
> > > > [3] Konen, Kai, et al. "Style Vectors for Steering Generative Large Language Model." arXiv preprint arXiv:2402.01618 (2024).

---

> ### Author Response · Authors · 2024-12-02
>
> Dear Reviewer XAGT,
>
> Thank you for your comment and suggestions. Since the discussion period is ending on Dec 2nd, we kindly ask you to review our response to your concerns. We hope that our response could solve your questions, and we would greatly appreciate it if you could reconsider your score. Thank you again for your time!
>
> Authors

---

> ### Comment · Reviewer_XAGT · 2024-12-02
>
> Thank you for your thoughtful rebuttal responses, and apologies for my delay in submitting a response. I have increased my score.
>
> Overall I've increased my score for three primary reasons.
> 1) The additional baselines make it more clear that the proposed method works well, and increases my confidence in the underlying method. While the baselines are not directly comparable due to having access to different amounts of information, including them is useful to demonstrate to the reader that the proposed method is able to find useful steering directions.
> 2) I had not previously clearly understood the benefits for being able to learn steering vectors from user's implicit activity traces, which is a useful contribution on its own.
> 3) I had not previously understood the conciseness, toxicity, helpfulness, and humor results, which show that the proposed method is substantially more general than topic modeling.
>
> I still have the reservation that the theory does not very clearly map to the implementation details, and that this limitation is not clearly discussed in the paper.

---

> > ### Author Response · Authors · 2024-12-03
> >
> > Thank you for your comments and raising of score!

---

### Official Review · Reviewer_okhS · 2024-11-04

**Soundness:** 2
**Presentation:** 3
**Contribution:** 3
**Rating:** 6
**Confidence:** 2

**Summary:**

This paper introduces a novel model steering approach, CONFST, to enhance LLMs by aligning outputs more closely with user preferences. The authors provide a theoretical framework for understanding and quantifying model steering, a method that adjusts LLM outputs to meet diverse user needs without explicit instructions or exhaustive fine-tuning. CONFST operates by creating a “confident direction” based on user history, which is integrated into LLM activations to guide outputs toward desired styles or topics. Unlike existing bidirectional steering methods, CONFST supports multiple preferences, making it a scalable solution for complex user alignment tasks. Validation experiments on models like GPT-2 XL and Mistral demonstrate CONFST's effectiveness, particularly in topic and style shifts.

**Strengths:**

S1. The paper provides a clear theoretical basis for understanding the mechanics of model steering, which is often underexplored in existing literature.
S2. CONFST’s ability to handle multiple alignment directions and user preferences offers a versatile approach, going beyond simple bidirectional steering.
S3. By reducing the need for fine-tuning, CONFST presents a resource-efficient alternative for LLM personalization, which is particularly valuable for large-scale models.

**Weaknesses:**

W1. The examples in Figure 1 are not intuitive. Why does a user hide her need when using LLMs? How can the proposed method successfully guess what a user thinks?
W2. Despite the theoretical foundation, how specific steering directions map to nuanced user preferences could benefit from further empirical examples or interpretability measures.
W3. The success of CONFST appears sensitive to confidence threshold settings, which may require careful tuning, especially in applications with noisy data.

**Questions:**

Please refer to Weaknesses.

---

> ### Author Response · Authors · 2024-11-22
> **Response to Reviewer okhS**
>
> >> Weakness 1: The examples in Figure 1 are not intuitive. Why does a user hide her need when using LLMs? How can the proposed method successfully guess what a user thinks?
>
> **Response**: We believe that the example in Figure 1 is **intuitive** can reflect the **realistic** situation. In many cases, users interacting with LLMs fail to explicitly convey their **implicit** preferences or their preference can't be expressed accurately. See similar settings in other papers. For example, in [1], user’s dialogue history are used to automatically build implicit user profiles, which guides the output; in [2], LLM commnicates with the user to collect information and infer the interest and preference of user.
>
> As a specific use case of our method, consider this example: A user asks an LLM, "What is a shrimp dish that is keto-friendly?" The user, being skilled in cooking, only wants the dish name and not detailed cooking instructions. However, they might not explicitly mention this preference, as it may not occur to them that the prompt could be interpreted differently—for example, as a request for a full recipe. By default, the LLM generates a response that includes detailed cooking instructions along with the dish name, which does not align with the user’s intent. While the user could refine their prompt to clarify their preference, this additional step reduces the efficiency of the interaction.
>
> If an LLM fails to generate a response that meets user expectations, users must provide explicit feedback to guide the LLM in adjusting the output to align with their preferences. Our method eliminates the need for collecting explicit user instructions on preferred generation styles or content. Instead, it learns and understands users' expectations by analyzing their past interactive history, enabling the LLM to align with users’ implicit preferences more effectively.
>
>
> >>Weakness 2:  Despite the theoretical foundation, how specific steering directions map to nuanced user preferences could benefit from further empirical examples or interpretability measures.
>
> **Response**: Thanks for your suggestion. We have already provided empirical visualization of steering directions by PCA analysis in **Appendix C Fig. 11 and Fig.12**, as an interpretable measure. In Figure 11, we use AgNews dataset as input, and get the activations of each sample from the first layer. Then we map each high dimension activation into 2-dimension space and make the scatterplot. The figure on the left illustrates the case where there are 4 preferences (each represented by one class of news), and the different preferences are difficult to be separated due to high data noise, which can leads to  the low occurence probability of ground-truth $P(\theta^*|f(\mathcal{T}(A_{1:n})))$. On the right side, we use the CONFST algorithm to select the confident steering directions, where the 4 preferences are more separable, thus the inference of latent user preference $\theta^*$ from steering direction can be more accurate. In Fig.12, we use the conciseness steering as an example. We provide an additional visualization in the form of a PCA plot, illustrating the activations of the last token for concise responses versus long responses in Fig.12 Appendix C. We can conclude from Fig. 12 that we can clearly see the difference between concise and long responses, while our CONFST algorithm selects the circled confident conciseness directions to construct steering vector. These selected directions typically correspond to responses with a simple "Yes" or "No" for yes/no questions, or a single word for "what" questions.Fig. 11 clearly illustrates an interpretable example to show the theoretical connection between steering direction and user preference.

---

> > ### Author Response · Authors · 2024-11-22
> > **Response to Reviewer okhS (continue)**
> >
> > >>Weakness 3: The success of CONFST appears sensitive to confidence threshold settings, which may require careful tuning, especially in applications with noisy data.
> >
> > **Response**: Thanks for your comment. We acknowledge that parameter tuning is necessary for our experiments, but it is also standard in the model steering literature. For instance, prior work such as [3] emphasizes the importance of tuning the magnitude of the steering direction to achieve optimal steering effects. Additionally, tuning the confidence level is not arbitrary; it follows a specific pattern similar as [3]: Setting the threshold too low can introduce noise, while setting it too high limits the sample size for steering. To find an optimal threshold, we can simply employ the grid search, which is a standard parameter tuning method. For example, in searching for the conciseness direction with HelpSteer2 dataset, the final threshold is set to be $0.98$, consisting of $43$ selected directions. We set it initially as $0.8$, and find the generated output is still long because it contains directions irrelevant to conciseness, so we gradually increase the threshold until $0.98$, when we observe the generated content is significantly shorter.
> >
> >
> >
> > [1] Qian, Hongjin, et al. "Learning implicit user profile for personalized retrieval-based chatbot." proceedings of the 30th ACM international conference on Information & Knowledge Management. 2021.
> >
> > [2] Hu, Chenhao, et al. "Learning to infer user implicit preference in conversational recommendation." Proceedings of the 45th International ACM SIGIR conference on research and development in information retrieval. 2022.
> >
> > [3] Li, Kenneth, et al. "Inference-time intervention: Eliciting truthful answers from a language model." Advances in Neural Information Processing Systems 36 (2024).

---

### Author Response · Authors · 2024-11-22
**General response to all reviewers**

**General response**:
We thank all the reviewers for the valuable comments and suggestions. During the rebuttal phase, we:

(1) Add two additional baselines Activation Addition and In-context learning in Fig.4 and Fig.5 despite these two baselines are not directly comparable to our method. The detailed setting are addressed in responses to individual reviewers.

(2) Correct the typos in texts and figures.

(3) Provide the detailed parameters in Appendix E.

Regarding the weaknesses and questions, please refer to our detailed responses to individual reviewers for specific clarifications and explanations.

---

> ### Author Response · Authors · 2024-11-25
> **Discussion on rebuttal and score**
>
> Dear reviewers,
>
> The end of author-reviewer discussion phase is approaching and we have addressed all the questions in the rebuttal. If the responses solve your concern, please consider raising the score. And we are happy to further discuss about our paper.
>
> Thanks,
> Authors

---

### Author Response · Authors · 2024-12-04

Dear Reviewers,

We appreciate your efforts in reviewing and giving suggestions to our work. We have addressed and responded to all the valuable concerns and questions. Since the discussion period is ending soon, we sincerely ask you to read our response and reconsider the scores. To summarize our work during the discussion phase, we have:

(1) Add two additional baselines Activation Addition and In-context learning in Fig.4 and Fig.5 despite these two baselines are not directly comparable to our method. The detailed setting are addressed in responses to individual reviewers.

(2) Correct the typos in texts and figures.

(3) Provide the detailed parameters in Appendix E.

(4) Corrected the typos in the revision.

We would appreciate it if you could take our additional experiments and author responses into consideration for the decision!

Authors

---

### Meta-Review · Area_Chair_CefY · 2024-12-21

**Metareview:**

This paper introduces CONFST, a novel model steering approach designed to enhance large language models (LLMs) by aligning their outputs more effectively with user preferences. The authors present a theoretical framework for understanding and quantifying model steering, a technique that adjusts LLM outputs to meet diverse user needs without requiring explicit instructions or extensive fine-tuning. CONFST works by constructing a "confident direction" based on user history, which is then integrated into the model’s activations to guide outputs toward specific styles or topics. Unlike traditional bidirectional steering methods, CONFST supports multiple user preferences, offering a scalable solution for complex alignment tasks. Experiments conducted on models such as GPT-2 XL and Mistral validate the effectiveness of CONFST, particularly in managing topic and style shifts.

Strong points:
+ The theoretical analysis is nice.
+ The idea of the paper is novel

Negative points:
- The evaluation metrics are not appropriate
- The datasets are not comprehensive
- The baselines are not sufficient

**Additional Comments On Reviewer Discussion:**

In the rebuttal period, the authors provided detailed responses. However, some concerns of reviewer wHLb are not addressed. In addition, the paper ratings are 5,5,6,6, which are not strong enough to achieve acceptance.

---

### Decision · Program_Chairs · 2025-01-22

Reject